# Deep Temporal Clustering: Fully unsupervised learning of time-domain features

## Abstract

Unsupervised learning of time series data, also known as temporal clustering, is a challenging problem in machine learning. Here we propose a novel algorithm, Deep Temporal Clustering (DTC), to naturally integrate dimensionality reduction and temporal clustering into a single end-to-end learning framework, fully unsupervised. The algorithm utilizes an autoencoder for temporal dimensionality reduction and a novel temporal clustering layer for cluster assignment. Then it jointly optimizes the clustering objective and the dimensionality reduction objective. Based on requirement and application, the temporal clustering layer can be customized with any temporal similarity metric. Several similarity metrics and state-of-the-art algorithms are considered and compared. To gain insight into temporal features that the network has learned for its clustering, we apply a visualization method that generates a region of interest heatmap for the time series. The viability of the algorithm is demonstrated using time series data from diverse domains, ranging from earthquakes to spacecraft sensor data. In each case, we show that the proposed algorithm outperforms traditional methods. The superior performance is attributed to the fully integrated temporal dimensionality reduction and clustering criterion.

## 1 Introduction

Deep learning has become the dominant approach to supervised learning of labeled data (LeCun et al., 2015; Schmidhuber, 2015). However, in many applications the data labels may not be available or be reliable. A variety of techniques have been developed for unsupervised learning where the algorithms draw inferences from unlabeled data. However, the progress in learning of complex structure in the data has so far been largely restricted to labeled datasets, while relatively little attention was paid to learning of complex, high-level structure and features of unlabeled data. The standard unsupervised techniques include clustering approaches which organize similar objects into clusters. Such techniques differ in the method for organizing the data as well as the metrics to measure similarity. While clustering techniques have been successfully applied to static data, their extension to time series data remains an open problem. This has left a gap in technology for accurate unsupervised learning of time series data which encompass many areas of science and engineering such as financial trading, medical monitoring, and event detection (Aghabozorgi et al., 2015).

The problem of unsupervised time series clustering is particularly challenging. Time series data from different domains exhibit considerable variations in important properties and features, temporal scales, and dimensionality. Further, time series data from real world applications often have temporal gaps as well as high frequency noise due to the data acquisition method and/or the inherent nature of the data (Antunes & Oliveira, 2001).

To address the issues and limitations of using standard clustering techniques on time series data, we present a novel algorithm called deep temporal clustering (DTC). A key element of DTC is the transformation of the time series data into a low dimensional latent space using a trainable network, which here we choose to be a deep autoencoder network that is fully integrated with a novel temporal clustering layer. The overview of our method is illustrated in Figure 1. The latent representation is compatible with any temporal similarity metric.

The proposed DTC algorithm was designed based on the observation that time series data have informative features on all time scales. To disentangle the data manifolds, i.e., to uncover the latent

dimension(s) along which the temporal or spatio-temporal unlabeled data split into two or more classes, we propose the following three-level approach. The first level, implemented as a CNN, reduces the data dimensionality and learns the dominant short-time-scale waveforms. The second level, implemented as a BI-LSTM, reduces the data dimensionality further and learns the temporal connections between waveforms across all time scales. The third level performs non-parametric clustering of BI-LSTM latent representations, finding one or more spatio-temporal dimensions along which the data split into two or more classes. The unique ability of this approach to untangle the data manifolds without discarding the information provided by the time course of the data (e.g., in contrast to PCA-based methods) allows our approach to achieve high performance on a variety of real-life and benchmark datasets without any parameter adjustment.

DTC also includes an algorithm to visualize the cluster-assignment activations across time, a feature not available in traditional clustering algorithms. This allows the localization of events in unlabeled time series data, and provides explanation (as opposed to black-box approaches) regarding the most informative data features for class assignment.

To the best of our knowledge, this is the first work on the application of deep learning in temporal clustering. The main contribution of our study is the formulation of an end-to-end deep learning algorithm that implements objective formulation to achieve meaningful temporal clustering. We carefully formulated the objective to encompass two crucial aspects essential for high clustering accuracy: an effective latent representation and a similarity metric which can be integrated into the learning structure. We demonstrate that the end-to-end optimization of our network for both reconstruction loss and clustering loss offers superior performance compared to cases where these two objectives are optimized separately. We also show that DTC outperforms current state-of-the-art, k-Shape (Paparrizos & Gravano, 2015) and hierarchical clustering with complete linkage, when evaluated on various real world time series datasets.

## 2   RELATED WORK

Much of the existing research in temporal clustering methods has focused on addressing one of two core issues: an effective dimensionality reduction and choosing an appropriate similarity metric.

One class of solutions use application dependent dimensionality reduction to filter out high frequency noise. Examples include adaptive piecewise constant approximation (Keogh et al., 2001) and nonnegative matrix factorization (Brockwell & Davis, 2013). One drawback of such approaches is that the dimensionality reduction is performed independent of the clustering criterion, resulting in the potential loss of long range temporal correlations as well as filtering out of relevant features. Other limitations include hand crafted and application specific nature of the transformations which require extensive domain knowledge of the data source and are difficult to design.

A second class of solutions have focused on the creation of a suitable similarity measure between two time series by taking into account features such as complexity (Batista et al., 2011), correlation (Galeano & Peña, 2000; Golay et al., 1998), and time warping (Berndt & Clifford, 1994). These similarity measures were then incorporated into traditional clustering algorithms such as k-means or hierarchical clustering. Montero et al. (2014) conducted a detailed study on various similarity metrics and showed that the choice of similarity measure has a significant impact on the results. However, a good similarity measure, in the absence of proper dimensionality reduction, may not be sufficient to obtain optimal clustering results due to the complexity and high dimensional nature of time series data.

The studies mentioned have shown that casting the original time series data into a low dimensional latent space is well suited for temporal clustering. These approaches lack a general methodology for the selection of an effective latent space that captures the properties of time series data. Another key step in achieving meaningful clustering results is ensuring the similarity metric is compatible with the temporal feature space.

Recent research in clustering methods for static data achieved superior performance by jointly optimizing a stacked autoencoder for dimensionality reduction and a k-means objective for clustering (Xie et al. (2016), Yang et al. (2016)). However, those approaches were designed for static data, and are not well suited for time series data clustering.

## 3 PROPOSED METHOD: DEEP TEMPORAL CLUSTERING

Consider $n$ unlabeled instances, $\mathbf{x}_1, ..., \mathbf{x}_n$, of a temporal sequence $\mathbf{x}$. The goal is to perform unsupervised clustering of these $n$ unlabeled sequences into $k \leq n$ clusters, based on the latent high-level features of $\mathbf{x}$.

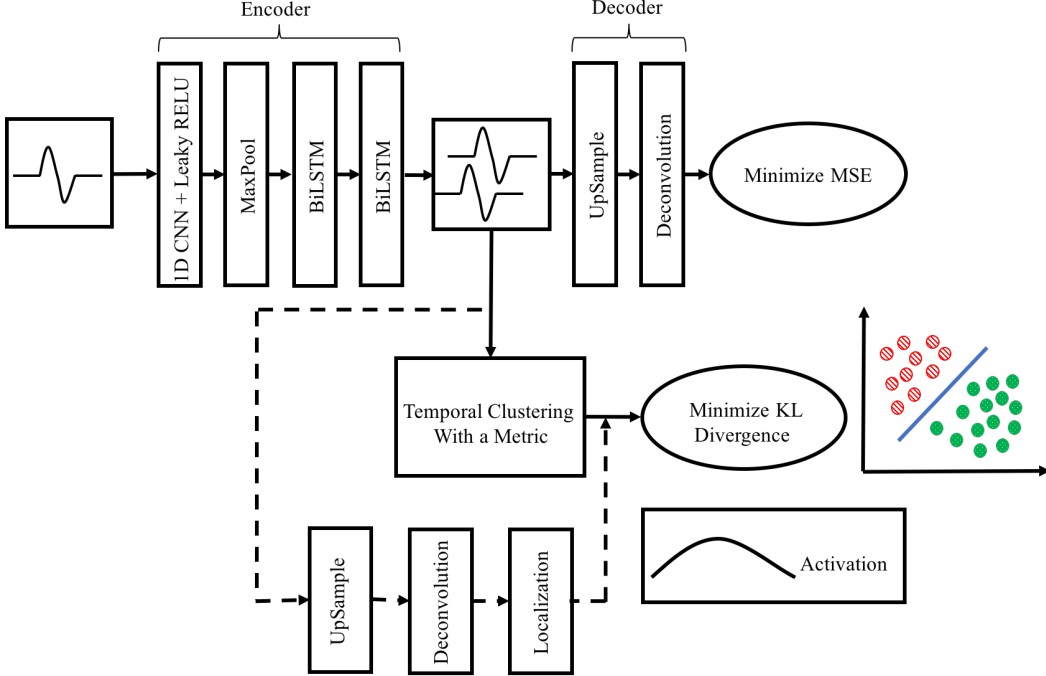

Figure 1: The overview of proposed DTC. The input signal is encoded into a latent space by a convolutional autoencoder followed by a BI-LSTM, as described in the Section 3.1. The convolutional autoencoder and the BI-LSTM constitute a temporal autoencoder (TAE). The latent representation of the BI-LSTM is then fed to a temporal clustering layer (Section 3.2), generating the cluster assignments. The dashed lines indicate the heatmap generating network (Section 3.4).

### 3.1 EFFECTIVE LATENT REPRESENTATION

Effective latent representation is a key aspect of the temporal clustering. We achieve this by making use of a temporal autoencoder (TAE) as shown in Figure 1. The first level of the network architecture consists of a 1D convolution layer, which extracts key short-term features (waveforms), followed by max pooling layer of size $P$. Leaky rectifying linear units (L-ReLUs) are used. The first level thus casts the time series into a more compact representation while retaining most of the relevant information. This dimensionality reduction is crucial for further processing to avoid very long sequences which can lead to poor performance. First-level activations are then fed to the second level (Bidirectional LSTM) to obtain the latent representation. We use BI-LSTM to learn temporal changes in both time directions. This allows to collapse the input sequences in all dimensions except temporal, and to cast the input into a much smaller latent space. Finally, the clustering layer assigns the BI-LSTM latent representation of sequences $\mathbf{x}_i$, $i = 1...n$, to clusters. Learning in both 1D CNN and BI-LSTM is driven by interleaved minimization of two cost functions. First cost function is provided by the mean square error (MSE) of the input sequence reconstruction from the BI-LSTM latent representation; this ensures that the sequence is still well represented after the dimensionality reduction in levels 1 and 2. Reconstruction is provided by an upsampling layer of size $P$ followed by a deconvolutional layer to obtain autoencoder output. The second cost function is provided by the clustering metric (e.g. KL divergence, see below) of level 3; this ensures that the high-level features that define the subspace spanned by the cluster centroids indeed separate the sequences $\mathbf{x}_i$, $i = 1...n$ into $k$ clusters of distinct spatio-temporal behavior. The clustering metric optimization modifies the weights in the BI-LSTM and in the CNN. As the result, the high-level features encoded by

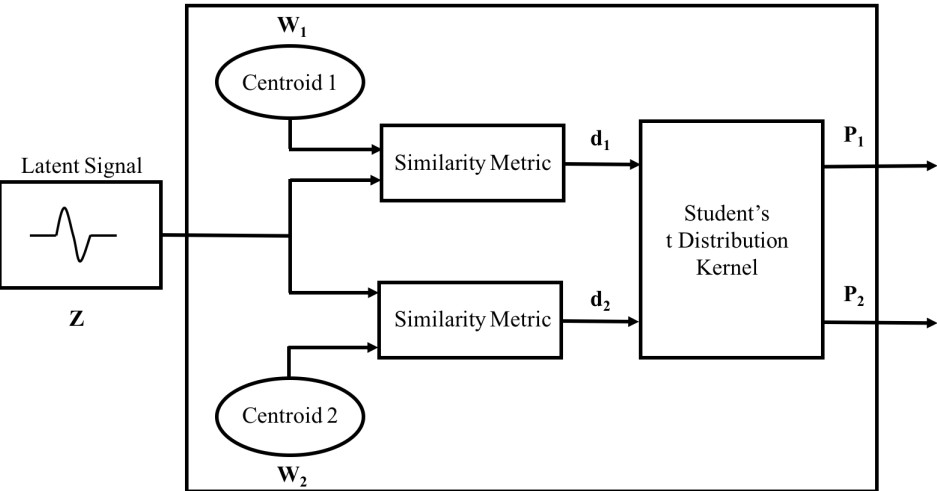

Figure 2: Temporal clustering layer for a 2 cluster problem.

the BI-LSTM optimally separate the sequences into clusters, thus disentangling the spatio-temporal manifolds of the $\mathbf{x}$ dynamics.

We emphasize that the end-to-end optimization of our network for both objectives (reconstruction loss and clustering loss) efficiently extracts spatio-temporal features that are best suited to separate the input sequences into categories (clusters), that is, to disnetangle the complicated high-dimensional manifolds of input dynamics. This in contrast to the traditional approaches where dimensionality reduction (e.g., truncated PCA or stacked autoencoders) only optimizes reconstruction, while clustering only optimizes separation. This results, in the traditional approaches, in separation being carried out in the space of latent features that are not best suited to make the data more separable. Direct comparison (see Section 4) shows marked improvement in unsupervised categorization when using end-to-end optimization relative to disjointly optimized dimensionality reduction and clustering. Forgoing initial dimensionality reduction and applying the traditional clustering approaches (e.g., k-means, dbscan, t-sne) directly to the spatio-temporal data $\mathbf{x}$ usually results in severe overfitting and poor performance. We further emphasize that our approach not only provides the effective end-to-end optimization, but also makes use of temporal continuity of the spatio-temporal data $\mathbf{x}$ to extract and encode the informative features on all time scales in the latent representation of the BI-LSTM.

## 3.2 TEMPORAL CLUSTERING LAYER

The temporal clustering layer consists of $k$ centroids $w_j, j \in 1..k$. To initialize these cluster centroids, we use the latent signals $z_i$ obtained by feeding input $\mathbf{x}_i$ through the initialized TAE. $z_i$ are then used to perform hierarchical clustering with complete linkage in the feature space Z through a similarity metric (discussed in the next section 3.2.2). We perform $k$ cut to obtain the clusters and then average the elements in each cluster to get initial centroids estimates $w_j, j = 1...k$.

### 3.2.1 CLUSTERING CRITERION

After we obtain an initial estimate of the centroids $w_j$, we train the temporal clustering layer using an unsupervised algorithm that alternates between two steps.

1. First, we compute the probability of assignment of input $\mathbf{x}_i$ belonging to the cluster $j$. The closer the latent representation $z_i$ of input $\mathbf{x}_i$ is to the centroid $w_j$, the higher the probability of $\mathbf{x}_i$ belonging to cluster $j$

2. Second, we update the centroids by using a loss function, which maximizes the high confidence assignments using a target distribution $p$, eq. 5, discussed in subsequent sections.

### 3.2.2 CLUSTERING ASSIGNMENT

When an input $z_i$ is fed to the temporal clustering layer, we compute distances $d_{ij}$ from each of the centroids $w_j$ using a similarity metric. Then we normalize the distances $d_{ij}$ into probability assignments using a Student's t distribution kernel (Maaten & Hinton, 2008). The probability assignment of latent signal belonging to $k^{th}$ cluster is as follows:

$$q_{ij} = \frac{\left(1 + \frac{siml(z_i, w_j)}{\alpha}\right)^{-\frac{\alpha+1}{2}}}{\sum_{j=1}^{k} \left(1 + \frac{siml(z_i, w_j)}{\alpha}\right)^{-\frac{\alpha+1}{2}}} \tag{1}$$

Here $q_{ij}$ is the probability of input $i$ belonging to cluster $j$, $z_i$ corresponds to the signal in the latent space Z, obtained from temporal autoencoder after encoding the input signal $\mathbf{x}_i \in X$. The parameter $\alpha$ is the number of degrees of freedom of the Students t distribution. In an unsupervised setting we can set $\alpha = 1$ as suggested by (Maaten & Hinton, 2008). Lastly $siml()$ is the temporal similarity metric which is used to compute the distance between the encoded signal $z_i$ and centroid $w_j$. We illustrate a 2 cluster example in Figure 2, where the latent input z is used to compute distances $d_1$ and $d_2$ from centroids $w_1$ and $w_2$ using a similarity metric, Later converted into probabilities $p_1$ and $p_2$ using a Students t distribution kernel.

In this study, we experiment with various similarity metrics as follows:

1. **Complexity Invariant Similarity(CID)** proposed by Batista et al. (2011) will compute the similarity based on the euclidean distance $ED$ which is corrected by complexity estimation $CF$ of the two series $x, y$. This distance is calculated as follows:

$$(x, y) = ED(x, y)CF(x, y) \tag{2}$$

where $CF(x, y)$ is a complexity factor defined as follows: $\frac{max(CE(x), CE(y))}{min(CE(x), CE(y))}$ and $CE(x)$ and $CE(y)$ are the complexity estimates of a time series $x$ and $y$, respectively. The core idea of CID is that with increase in the complexity differences between the series, the distance increases. Further, if both input sequences have the same complexity then the distance simply is the euclidean distance. The complexity of each sequence is defined as:

$$CE(x) = \sqrt{\sum_{t=1}^{N-1} (x_{t+1} - x_t)^2} \tag{3}$$

   where $N$ is length of the sequence

2. **Correlation based Similarity(COR)** as used by Golay et al. (1998), computes similarities using the estimated pearsons correlation $\rho$ between the latent representation $z_i$ and the centroids $w_j$. In this study we compute the COR as follows:

$$COR = \sqrt{2(1 - \rho)} \tag{4}$$

   where $\rho$ is the pearson's correlation, given by $\rho_{x,y} = \frac{\text{cov}(x,y)}{\sigma_x \sigma_y}$ and $cov$ is the covariance.

3. **Auto Correlation based Similarity(ACF)** as used in Galeano & Peña (2000), computes the similarity between the latent representation $z_i$ and the centroids $w_j$ using autocorrelation (ACF) coefficients and then performs weighted euclidean distance between the auto-correlation coefficients.

4. **Euclidean(EUCL)** is the euclidean distance between the input signal and each of the centroids.

### 3.2.3 CLUSTERING LOSS COMPUTATION

To train the temporal clustering layer iteratively we formulate the objective as minimizing the KL divergence loss between $q_{ij}$ and a target distribution $p_{ij}$. Choice of $p$ is crucial because it needs

to strengthen high confidence predictions and normalize the losses in order to prevent distortion of latent representation. This is realized using

$$p_{ij} = \frac{q_{ij}^2/f_j}{\sum_{j=1}^{k} q_{ij}^2/f_j} \tag{5}$$

where $f_j = \sum_{i=1}^{n} q_{ij}$. Further empirical properties of this distribution were discussed in detail in Hinton et al. (2006) and Xie et al. (2016). Now using this target distribution, we compute the KL divergence loss:

$$L = \sum_{i=1}^{n} \sum_{j=1}^{k} p_{ij} log \frac{p_{ij}}{q_{ij}} \tag{6}$$

where $n$ and $k$ are number of samples in dataset and number of clusters respectively.

### 3.3 DTC Optimization

We perform batch-wise joint optimization of clustering and auto encoder by minimizing KL divergence loss and mean squared error loss, respectively. This optimization problem is challenging, and an effective initialization of the cluster centroids is highly important. Cluster centroids reflect the latent representation of the data. To make sure initial centroids represent the data well, we first pretrain parameters of the autoencoder to start with a meaningful latent representation. After pretraining, the cluster centers are then initialized by hierarchical clustering with complete linkage on embedded features of all datapoints. Later we update autoencoder weights and cluster centers using gradients $\frac{dL_c}{dz_i}$ and $\frac{dL_{ae}}{dz}$, see eqs. 7 and 8 below, using backpropagation mini-batch SGD. We also update target distribution $p$ during every SGD update using eq. 5.

$$\frac{dL_c}{dw_i} = \frac{\alpha}{1+\alpha} \sum_{j} \left(1 + \frac{siml(z_i, w_i)}{\alpha}\right) * (p_{ij} - q_{ij}) \frac{d\left(siml(z_i, w_i)\right)}{dw_i} \tag{7}$$

$$\frac{dL_{ae}}{dz} = \frac{d\left(\frac{1}{2}||x - x'||_2^2\right)}{dz} \tag{8}$$

Similar approaches have been used in Xie et al. (2016) and Yang et al. (2016). This helps prevent any problematic solutions from drifting too far away from the original input signal. As reconstruction of the original signal (reconstruction MSE reduction) is a part of the objective, the latent representation will converge at a suitable representation so as to minimize both the clustering loss and the MSE loss.

### 3.4 Visualizing The Heatmap

In most applications it is important to identify and localize the main data feature(s) contributing to the final classification. To do so, we create a heatmap-generating network following the approach used in (Hwang & Kim, 2016) to localize tumors in medical images using only image-level labels. Briefly, we use the cluster labels generated by our DTC network to train a new supervised hierarchical convolutional network to classify the inputs **x**. Activity in this new network allows us to generate heatmaps showing the parts of spatio-temporal inputs **x** that are most relevant for assigning inputs to clusters. Example of such heatmap for NASA data (see Section 4) is shown in Figure 3.

## 4 Experiments

Our networks were implemented and tested using Python, TensorFlow 1.3 and Keras 2.0 software on Nvidia GTX 1080Ti graphics processor. For the baseline algorithms the TSclust package (Montero et al., 2014) R implementations were used except for k-Shape where Python implementation[1] was used.

---

[1] https://github.com/Mic92/kshape.git

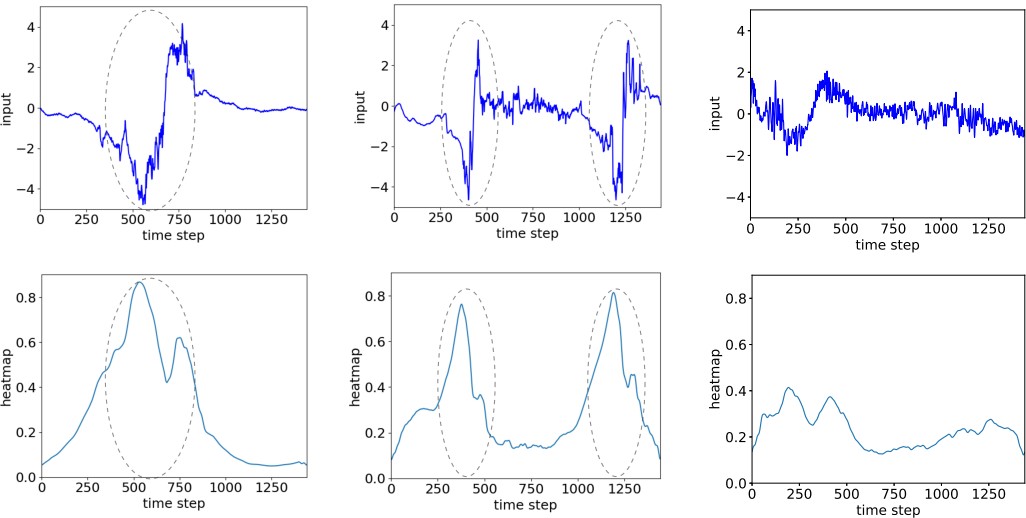

Figure 3: Localization of the event by the heatmap. The top panel in each column represents the input time series, where examples with one event (left panel), two events (center panel), and non-event (right panel) are shown. The events have a bipolar signature. DTC network learns to separate sequences containing events from non-event sequences unsupervised. The bottom panels show the corresponding heatmaps. Higher value in a heatmap corresponds to higher likelihood of event localization. The heatmap is found to correctly mark the time location of events. In the case of non-event (right panels), the heatmap amplitude is low as expected.

## 4.1 DATASETS

The performance of our DTC algorithm is evaluated on a wide range of real world datasets.

- We make use of a few publicly available UCR Time series Classification Archive datasets (Chen et al., 2015). Table 1 includes the properties of these datasets: the number of samples $N$, the time steps in each sequence $L$, and the class distribution ratio $r$. Since we are using these datasets as unlabeled data, we combine the training and test datasets for all UCR datasets for all experiments in this study.

- In addition, we use spacecraft magnetometer data from the recent NASA Magnetospheric Multiscale (MMS) Mission. The magnetospheric plasma environment exhibits many transient structures and waves. Here we are interested in automated detection of spacecraft crossings of the so-called flux transfer events (FTEs) in an otherwise turbulent medium. FTEs (Russell & Elphic, 1979) are characterized by bipolar signature in the $B_N$ component of the magnetic field. The $B_N$ data contains a total of 104 time series, with each sequence having 1440 time steps.

## 4.2 BASELINE METHODS

We compare the results of our DTC algorithm against two clustering methods, the hierarchical clustering with complete linkage, and k-Shape (Paparrizos & Gravano, 2015). k-Shape is the current state-of-the-art temporal clustering algorithm. It is a partitional clustering approach which preserves the shapes of time series and computes centroids under the scale and shift invariance.

We have considered four similarity metrics in our experiments: 1) *Complexity Invariant Distance(CID)*, 2) *Correlation based Similarity(COR)*, 3) *Auto Correlation based Similarity(ACF)*, and 4) *Euclidean Based Similarity(EUCL)*.

Table 1: The performance of our algorithm compared to the baseline metrics. Each entry represents the AUC averaged over 5 trials. Baseline metrics AUC's are averaged over 10 runs using bootstrap sampling. The first column indicates dataset size $N$, time series length $L$ and class distribution $r = \frac{\text{no pos class}}{\text{no neg class}}$ and $P$ is the pooling size used for DTC.

| Dataset($N$,$L$,$r$,$P$) | ACF | DTC ACF | CID | DTC CID | COR | DTC COR | EUCL | DTC EUCL | kshape |
|---|---|---|---|---|---|---|---|---|---|
| NASA MMS (104,1140,1.21,10) | 0.51 | 0.59 | 0.85 | 0.93 | 0.65 | 0.67 | 0.56 | 0.69 | 0.61 |
| BeetleFly (40,512,1.00,8) | 0.55 | 0.69 | 0.8 | 0.892 | 0.55 | 0.584 | 0.55 | 0.606 | 0.65 |
| BirdChicken (40,512,1.00,8) | 0.7 | 0.792 | 0.6 | 0.732 | 0.55 | 0.712 | 0.55 | 0.772 | 0.52 |
| Computers (500,720,1.00,10) | 0.58 | 0.64 | 0.51 | 0.68 | 0.55 | 0.555 | 0.55 | 0.58 | 0.58 |
| Earthquakes (461,512,0.25,8) | 0.508 | 0.569 | 0.508 | 0.588 | 0.546 | 0.549 | 0.546 | 0.540 | 0.59 |
| MoteStrain (1272,84,0.86,4) | 0.68 | 0.89 | 0.57 | 0.81 | 0.60 | 0.93 | 0.60 | 0.93 | 0.88 |
| Phalanges OutlinesCorrect (2658,80,1.77,4) | 0.586 | 0.522 | 0.501 | 0.529 | 0.501 | 0.525 | 0.501 | 0.556 | 0.56 |
| ProximalPhalanx OutlineCorrect (891,80,2.12,4) | 0.52 | 0.678 | 0.5 | 0.66 | 0.52 | 0.62 | 0.52 | 0.63 | 0.65 |
| ShapeletSim (200,500,1.00,10) | 0.54 | 0.74 | 0.83 | 0.91 | 0.52 | 0.55 | 0.52 | 0.53 | 0.56 |
| SonyAIBO RobotSurfaceII (980,65,1.61,5) | 0.56 | 0.72 | 0.827 | 0.85 | 0.57 | 0.82 | 0.57 | 0.83 | 0.65 |
| SonyAIBO RobotSurface (621,70,0.78,5) | 0.74 | 0.94 | 0.51 | 0.81 | 0.58 | 0.78 | 0.58 | 0.66 | 0.74 |
| ItalyPower Demand (1096,24,1,4) | 0.59 | 0.63 | 0.60 | 0.66 | 0.54 | 0.57 | 0.54 | 0.61 | 0.52 |
| WormsTwoClass (258,900,1.37,10) | 0.53 | 0.62 | 0.59 | 0.61 | 0.55 | 0.56 | 0.55 | 0.51 | 0.55 |

### 4.3 EVALUATION METRICS

We have expert labels for our datasets. However, our entire training pipeline is unsupervised and we only used the labels to measure the performance of the models as a classifier. To this end, we use the Receiver Operating Characteristics (ROC) (Fawcett, 2006) and area under the curve (AUC) as our evaluation metrics. We used bootstrap sampling (Singh & Xie, 2008) and averaged the ROC curves over 5 trials.

### 4.4 IMPLEMENTATION AND PARAMETER INITIALIZATION

Parameter optimization using cross-validation is not feasible in unsupervised clustering. Hence we use commonly used parameters for DTC and avoid dataset specific tuning as much as possible. The convolution layer has 50 filters with kernel size 10 and two Bi-LSTM's have filters 50 and 1 respectively. The pooling size $P$ is chosen such that it makes the latent representation size $<$ 100 for faster experimentation. $P$ for each experiment is listed in the first column in Table 1. The deconvolutional layer has kernel size 10. We initialize all weights to a zero-mean Gaussian distribution with a standard deviation of 0.01. Autoencoder network is pre-trained, using the Adam

optimizer, over 10 epochs. Temporal clustering layer centroids are initialized using hierarchical clustering with complete linkage and using the chosen metric.

The entire deep architecture is jointly trained for clustering and auto encoder loss until convergence criterion of 0.1% change in cluster assignment is met. The mini-batch size is set to 64 for both pre-training and end-to-end fine tuning of the network and the starting learning rate is set to 0.1. These parameters are the same for all experiments and are held constant across all datasets. The baseline algorithms we use are parameter free.

### 4.5 QUALITATIVE ANALYSIS

We show in Fig. 3 results of DTC for three distinct time series from the MMS dataset. The top panels shows the three input time series. The bottom panels show the activation map as generated by the algorithm. Activation map profiles correlate well with the location of the bipolar signatures of the events. Events are highlighted by the dashed ovals for readers' convenience. Note that the second time series has two events and the heatmap identifies both of the events correctly. The third time series (right panels) is a non-event and the algorithm correctly identifies it as such (heatmap activation is low).

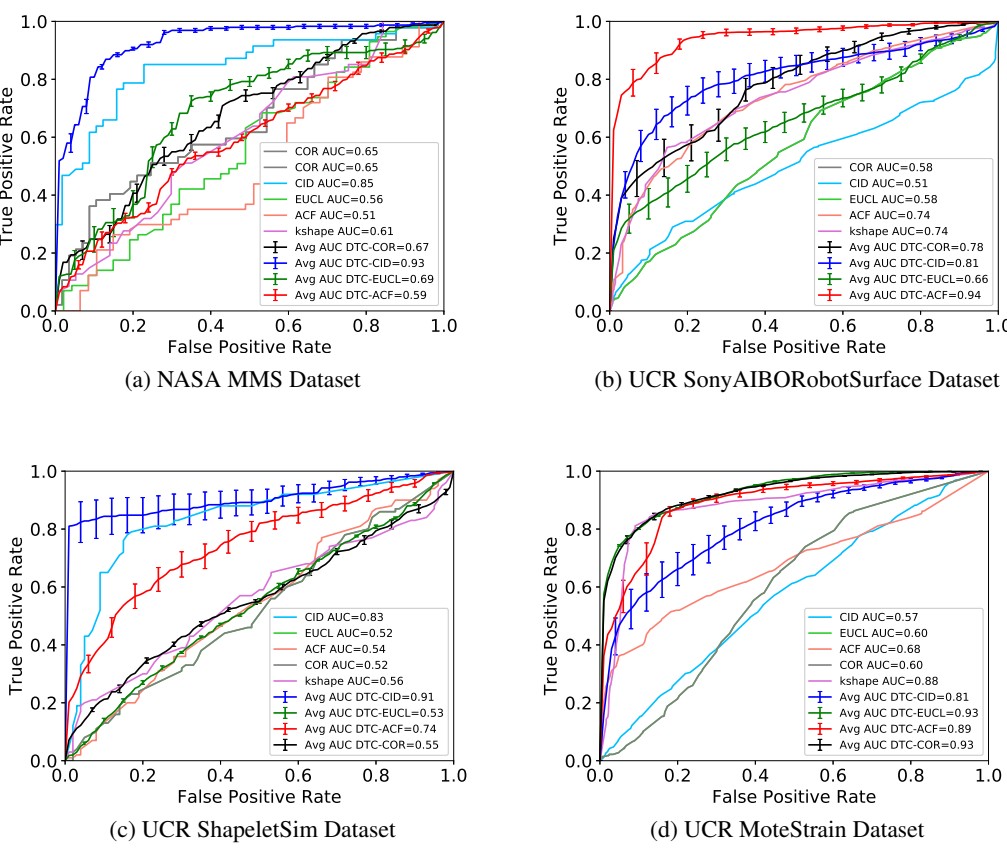

Figure 4: Performance of our algorithm compared to the baseline metrics. The mean ROC and standard error bars are shown.

### 4.6 QUANTITATIVE ANALYSIS

One of the main points of the paper is that the joint training of the two objectives, reconstruction loss and clustering loss, yields superior performance compared to the disjointed training where each loss is optimized independently. Direct comparison between joint end-to-end training of the DTC vs disjoint DTC training (training the temporal autoencoder to minimize reconstruction loss and then

training the clustering to minimize clustering loss) on the MMS dataset, all other network parameters being identical, results in the average AUC of 0.93 for joint training vs. average AUC of 0.88 for disjointed training.

Next, we include in Table 1 a detail comparison of results from DTC and the two baseline clustering techniques for 4 different similarity metrics and over 13 different datasets. We can see that our algorithm was able to improve the baseline performance in all of the datasets over all metrics. Although k-Shape performed well on a few datasets, chosen the right metric DTC outperforms k-Shape for all the datasets considered.

We also show the comparison of ROCs in Fig. 4. In order to keep the figure readable, we only show the ROCs for 4 datasets over 4 similarity metrics. These results illustrate DTC to be robust and provide superior performance over existing techniques across datasets from different domains, with different dataset sizes $N$, different lengths $L$, as well as a range of data imbalances denoted by $r$.

## 5 CONCLUSION

In this work we addressed the question of unsupervised learning of patterns in temporal sequences, unsupervised event detection and clustering. Post-hoc labeling of the clusters, and comparison to ground-truth labels not used in training, reveals high degree of agreement between our unsupervised clustering results and human-labeled categories, on several types of datasets. This indicates graceful dimensionality reduction from inputs with extensive and complex temporal structure to one or few-dimensional space spanned by the cluster centroids. As most natural stimuli are time-continuous and unlabeled, the approach promises to be of great utility in real-world applications. Generalization to multichannel spatio-temporal input is straightforward and has been carried out as well; it will be described in more detail in a separate paper.

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
