# OpenReview forum: "Deep Temporal Clustering: Fully unsupervised learning of time-domain features"
_ICLR.cc/2018/Conference — Reject_

### Official Review · AnonReviewer2 · 2017-11-25
**The authors proposed an algorithm named Deep Temporal Clustering (DTC) that integrates autoencoder with time-series data clustering. The idea is interesting, but there are many drawbacks of the paper, i.e. problem statement, comparison to state-of-the-art.**

**Rating:** 3
**Confidence:** 5

**Review:**

The authors proposed an algorithm named Deep Temporal Clustering (DTC) that integrates autoencoder with time-series data clustering. Compared to existing methods, DTC used a network structure (CNN + BiLSTM) that suits time-series data. In addition, a new clustering loss with different similarity measures are adopted to DTC. Experiments on different time-series data show the effectiveness of DTC compared to complete link.

Although the idea of applying deep learning for temporal clustering is novel and interesting, the optimization problem is not clearly stated and experiments section is not comprehensive enough.

Here are the detailed comments.
The methods are described in a higher level language. The formula of overall loss function and its optimization should be written down to avoid unclearness.
The framework adopt the K-medoid clustering idea. But complete-link is used for initialization and comparison. Is that a difference? In addition, how to generate K centroids from complete-link clustering is not described at all.
The author Dynamic Time Warping is too expensive to integrate into DTC. However, most of the evaluated dataset are with small time points. Even for the longer ones, DTC does dimensionality reduction to make the time-series shorter. I do not see why quadratic computation is a problem here. DTW is most effective similarity measure for time-series data clustering. There is no excuse to skip it.
Is DTC robust to hyperparameters? If not, are there any guidelines to tune the hyperparameters, which is very important for unsupervised clustering.

In summary, the method need to be described clearer, state-of-the-arts need to be compared and the usability of the method needs to be discussed. Therefore, at the current stage the paper cannot be accepted in my opinion.

---

### Official Review · AnonReviewer1 · 2017-11-27
**Deep Temporal Clustering**

**Rating:** 5
**Confidence:** 4

**Review:**

This paper proposes an algorithm for jointly performing dimensionality reduction and temporal clustering in a deep learning context.  An autoencoder is utilized for dimensionality reduction alongside a clustering objective - that is the autoencoder optimizes the mse (using LSTM layers are utilized in the autoencoder for modelling temporal information), while the latent space  is fed into the temporal clustering layer.  The clustering/autoencoder objectives are optimized in an alternating optimization fashion.

The main con lies in this work being very closely related to t-sne, i.e. compare the the temporal clustering loss based on kl-div (eq 6) to t-sne.  If we consider e.g., a linear 1-layer autoencoder to be equivalent to PCA (without the rnn layers), in essence this formulation is closely related to applying pca to reduce the initial dimensionality and then t-sne.

Also, do the cluster centroids appear to be roughly stable over many runs of the algorithm? As the authors mention, the method is sensitive to intitialization.  As the averaged results over 5 runs are shown, the standard deviation would be helpful towards showing this empirically.

On the positive side, it is likely that richer representations can be obtained via this architecture, and results appear to be good with comparison to other metrics

The section of the paper that discusses heat-maps should be written more clearly.  Figure 3 is commented with respect to detecting an event - non-event but the process itself is not clearly described as far as I can see.

minor note: the dynamic time warping is formally not a metric

---

### Official Review · AnonReviewer3 · 2017-11-28
**A paper with interesting ideas but lacking convincing evidence**

**Rating:** 4
**Confidence:** 4

**Review:**


Summary:
The authors proposed an unsupervised time series clustering methods built with deep neural networks. The proposed model is equipped with an encoder-decoder and a clustering model. First, the encoder employs CNN to shorten the time series and extract local temporal features, and the CNN is followed by bidirectional LSTMs to get the encoded representations. A temporal clustering model and a DCNN decoder are applied on the encoded representations and jointly trained. An additional heatmap generator component can be further included in the clustering model. The authors compared the proposed method with hierarchical clustering with 4 different temporal similarity methods on several univariate time series datasets.

Detailed comments:
The problem of unsupervised time series clustering is important and challenging. The idea of utilizing deep learning models to learn encoded representations for clustering is interesting and could be a promising solution.

One potential limitation of the proposed method is that it is only designed for univariate time series of the same temporal length, which limits the usage of this model in practice. In addition, given that the input has fixed length, clustering baselines for static data can be easily applied and should be compared to demonstrate the necessity of temporal clustering.

Some important details are missing or lack of explanations. For example, what is the size of each layer and the dimension of the encoded space? How much does the model shorten the input time series and how is this be determined?

How does the model combine the heatmap output (which is a sequence of the same length as the time series) and the clustering output (which is a vector of size K) in Figure 1? The heatmap shown in Figure 3 looks like the negation of the decoded output (i.e., lower value in time series -> higher value in heatmap). How do we interpret the generated heatmap?

From the experimental results, it is difficult to judge which method/metric is the best. For example, in Figure 4, all 4 DTC-methods achieved the best performance on one or two datasets. Though several datasets are evaluated in experiments, they are relatively small. Even the largest dataset (Phalanges OutlinesCorrect) has only 2 thousand samples, and the best performance is achieved by one of the baseline, with AUC score only 0.586 for binary classification.

Minor suggestion:
In Figure 3, instead of showing the decoded output (reconstruction), it may be more helpful to visualize the encoded time series since the clustering method is applied directly on those encoded representations.

---

### Decision · Program_Chairs · 2018-01-29
**ICLR 2018 Conference Acceptance Decision**

**Decision:**

Reject

**Comment:**

Joint optimization of dimensionality reduction and temporal clusters. Results suggest performance improvement in a variety of scenarios versus a baseline of a recent state-of-art clustering method.

Pro:
- Joint optimization may be new and results suggest performance improvement when done on NASA Magnetospheric Multiscale (MMS) Mission.

Con:
- Small datasets evaluated, impact unclear
- Breadth of possible applications unclear
- Similarities exist to prior works. Significance of novelty not clear.
- Unanimous consensus among reviewers that work is not in a state to be accepted.